# Dynamic Response Analysis of RC Frame against Progressive Collapse Based on Orthogonal Test

Changren Ke, Xianwei Li *[iD] and Junling Jiang

School of Civil Architecture and Environment, Hubei University of Technology, Wuhan 430068, China
* Correspondence: lxw1378045191@163.com; Tel.: +86-17340570491

**Abstract:** We aimed to investigate the extent to which the initial state generated by unexpected loads, such as explosions and impacts on the remaining structure, affects the dynamic response of the structure. The study used the orthogonal test method to obtain orthogonal table L25 (53) by combining five levels of each of the three factors of column removal time and initial velocity, and the initial displacement of the remaining structure under three failure scenarios of the corner, side, and internal columns at the ground floor of the reinforced concrete frame structure. The degree of influence of different factors on the structural dynamic response, and the result of the unifactorial impact of initial velocity and initial displacement of the remaining structure on the structural dynamic response in the case of failure of the bottom side columns, were obtained by the polar difference method. The results show that the analysis using the orthogonal test polar difference method revealed that the initial displacement has a more significant influence on the dynamic response of the structure, forming the main influencing factor. At the same time, the failure time and initial velocity have a smaller influence on the dynamic response of the structure as secondary influencing factors. In the case of unifactorial influence, the initial upward displacement and initial upward velocity are detrimental to the structure, leading to progressive collapse. In contrast, the initial downward velocity and initial downward displacement are favorable.

**Keywords:** progressive collapse; dynamic response; orthogonal test; initial condition; element failure time; structural failure; structural dynamics

## 1. Introduction

The collapse of the Ronan Point building in the UK in 1986 attracted worldwide attention. This accident led to research into the building structures' resistance to progressive collapse, with multiple corresponding code standards being proposed [1–3]. Progressive collapse is when a building structure is subjected to an accidental load, such as impact and blast load [4–11] or fires [12,13], that cause local damage, resulting in an unbalanced load. The remaining structure is unable to transmit the unbalanced load. As a result, adjacent elements break down sequentially, eventually leading to the collapse of the structure as a whole or causing damage that is out of proportion to the initial damage. The progressive collapse of the building structure under the action of an accidental load is a dynamic process. Scholars from various countries have conducted dynamic analyses of the structure to study the progressive collapse process under conditions closer to reality.

Tian, Orton et al. [14,15] performed a scale-down dynamic test of a reinforced concrete frame substructure that quickly removed the middle column. Liu et al. [16] established a two-beam single-column test model to study the dynamic response of the double-web angle steel connection joint after removing the column. Based on the reinforced concrete beam and column structure test model, Jun et al. [17] conducted an explosion demolition test of the middle column and compared it with the published quasi-static test results. Qian et al. [18,19] carried out model tests of the planar beam and column structure and spatial beam and column structure to study the influence of different structures, span lengths,

and a high span ratio on the internal force redistribution performance of the reinforced concrete structure under dynamic load. Pham et al. [20] established a reinforced concrete plane frame model, simulated the failure of the support column through a quick-release device, and carried out a series of dynamic tests, which showed the influence of inertia and strain rate on the structural response. Feng et al. [21] investigated the performance of different precast concrete frame structures with parameters such as connection detail, bar development length, beam section size, and rebar ratio to evaluate the load-carrying capacity, deformation capacity, and failure mode of the structure through four static tests. Based on this, dynamic tests were conducted to compare the progressive collapse resistance under static and dynamic loads and to discuss the effect of dynamic loads. Li et al. [22] developed a progressive collapse test apparatus for planar steel frames. Based on the measured deformations and strains, the device was used to study the dynamic response, collapse modes, and load transfer paths of three planar steel frames in the case of center column removal. The above scholars did not consider the initial condition of the remaining structure after column failure. However, the structure will have a certain initial velocity, initial displacement, and other initial conditions after being locally damaged by accidental loads. If these initial conditions are ignored, the dynamic response of the structure will be underestimated. Shi et al. [23] proposed a new method to continuously collapse the reinforced concrete frame structure by considering the adjacent components' initial conditions and damage under explosion load and establishing a three-layer, two-span supported concrete model to verify the method. The results show that, compared with the direct numerical simulation, this method requires less calculation and more accurately simulates the collapse process. Yu et al. [24] proposed a nonlinear single degree of freedom (SDOF) model, which used the Laplace transform technique to obtain the model closed analysis solution and verified it through dynamic experiments. The results showed that the deformation of the structure was underestimated when it showed upward displacement and initial velocity at the beginning of the progressive collapse. Qian et al. [25] established a numerical model through LS-DYNA to study the influence of the dynamic response of the reinforced concrete beam and plate substructure on the initial damage that occurred after the sudden removal of the middle column. The results showed that the damage caused by the sudden removal of the column reduced the progressive collapse resistance of the substructure.

Only single-factor analyses have been conducted on the residual structural dynamic response without studying the combined factor analysis and degree of influence of different factors on the structural dynamic response. In this paper, firstly, a dynamic response analysis of the initial displacement and initial velocity single factor was carried out under the working condition of side column failure. Then, five levels of initial velocity, initial displacement, and failure time were selected for each of the three working conditions of side column, corner column, and inner column failure by orthogonal test to arrange the combination reasonably. The data from the orthogonal tests were analyzed using the extreme difference method, and the most unfavorable combination could be derived from the maximum average degree of influence. The primary and secondary factors were derived from the magnitude of the extreme difference values. The orthogonal test method, also known as the orthogonal test design method, is widely used in chemical, mechanical, agricultural, and other fields [26,27]. The orthogonal test method is a scientific and reasonable arrangement of the multi-factor test method. When studying more complex problems, there will be multiple factors, and the different state of each factor is called the level. Various factors at numerous levels require an orthogonal test table to arrange the combination reasonably. The orthogonal table test arrangement scheme is representative and can fully reflect the influence of each factor on the index. Therefore, the orthogonal test method can reduce the number of tests. The test data obtained on this basis can analyze and draw correct conclusions consistent with the results obtained by comprehensive experiments. The orthogonal table is Ln(tq), where q represents the number of factors, t represents the number of levels, and n represents the number of trials. According to the orthogonal table

test protocol, the test results were obtained, and the average influence degree of each factor in the i-level state was calculated *ki*.

$$k_i = \sum_{j=1}^{n} s_{j_i} \tag{1}$$

(*Sji* is the maximum vertical displacement of a factor in the *j*th test at the i level.)

$$k_i = \frac{1}{n} K_{\cdot_j} \tag{2}$$

(n represents the number of levels.)

The extreme difference, *R*, for each factor, was obtained by the difference between the maximum value $k_{imax}$ and the minimum value $k_{imin}$ in each factor's average degree of influence in state *i*.

$$R = k_{imax} - k_{imin} \tag{3}$$

For each factor, a level of maximum average influence *ki* is chosen to determine the most unfavorable combination of factors that produce the maximum dynamic response of the structure. The extreme difference value *R* shows the degree of influence of each factor. The greater the extreme difference value, the greater the degree of influence of the factor, and the factors' order of priority can be derived.

## 2. Finite Element Analysis Model

### 2.1. Selection of Analytical Models

Referring to the actual engineering structure layout, a 6-layer and 4-span reinforced concrete space frame structure was designed by PKPM software as the research object [28]. The frame size was as follows:

The first-floor height was 4.9 m.

The remaining floor height was 3.3 m.

The span was 6 m.

The reinforced concrete beams and columns dimensions were 300 mm×, 500 mm and 500 mm × 500 mm.

C30 was selected for concrete.

The constant load of the frame floor was 5.0 KN/m$^2$.

The live floor load was 2.0 KN/m$^2$.

### 2.2. Fiber Beam Model

The reinforced concrete space frame was modeled and analyzed in this paper by ABAQUS finite element software. The calculation accuracy and efficiency were measured, and the three-dimensional, first order Timoskhenko beam element B31 in the ABAQUS unit library was used to simulate the beam and column members. The floor slab used the layered shell model S4R element, and the keyword *rebar command was entered in the inp file to insert the steel bar in the beam element.

### 2.3. Material Constitutive Selection

In this paper, iFberLUT, a steel and concrete structural fiber software based on the ABAQUS platform, was selected to select the iConcrete04 material model of ordinary concrete without considering the tensile, which considers the restraining effect of its stirrups on concrete by increasing the stress peak and its corresponding strain. The compression skeleton used the Kent–Park model Formula (4) modified by Scott [29]. The steel bar used

the iSteel01 material model, a double-fold line follow-up reinforcement model. The material parameters are shown in Tables 1 and 2.

$$\sigma = \begin{cases} Kf'_c\left[2\left(\frac{\varepsilon}{\varepsilon_0}\right) - \left(\frac{\varepsilon}{\varepsilon_0}\right)^2\right] (\varepsilon \le \varepsilon_0) \\ Kf'_c[1 - Z(\varepsilon - \varepsilon_0)] \ (\varepsilon_0 < \varepsilon \le \varepsilon_u) \\ 0.2Kf'_c \ (\varepsilon > \varepsilon_u) \end{cases} \quad (4)$$

**Table 1.** iConcrete04 material parameters.

| User Parameter | |
|---|---|
| | Mechanical constant |
| 1 | $2.75 \times 10^7$ |
| 2 | 0.0022 |
| 3 | $2.475 \times 10^7$ |
| 4 | 0.00778 |

**Table 2.** iSteel01 material parameters.

| User Parameter | |
|---|---|
| | Mechanical constant |
| 1 | $2.06 \times 10^{11}$ |
| 2 | $4.16 \times 10^8$ |
| 3 | 0.005 |

In the formula: K is the constraint enhancement coefficient; Z is the softening slope coefficient; $f'_c$ is the axial compressive strength of the cylinder; $\varepsilon_0$ is the strain corresponding to the compressive strength of concrete; $\varepsilon_u$ is the ultimate strain of concrete.

### 2.4. Rayleigh Damping

The self-resonance period of the overall structure is the main one, and the self-resonance period of the remaining structure is not very different from the self-oscillation period of the complete structure in the case of losing a single column. Therefore, this paper selected the self-resonance period under its first two orders' complete structure and circular frequency. $[C] = \alpha[M] + \beta[K]$ is the damping matrix, and its coefficients are calculated from the frequency of the first two circles. $\alpha$, $\beta$: the relationship between $\alpha$ and $\beta$ is $\xi_n = \frac{\alpha}{2\omega_n} + \frac{\beta\omega_n}{2}$. The value of $\xi$ is 0.05.

In the formula: $[M]$ is the mass matrix; $[K]$ is the stiffness matrix; $\xi$ is the damping ratio.

### 2.5. Example Verification

Yi, Wei-Jian [30] designed a 1:3 scale model of a one-bay reinforced concrete frame with beam and column sections of 100 mm × 200 mm and 200 × 200 mm, respectively, beam and column reinforcement of HRB400, hoop reinforcement of HPB235 and C30 concrete. The results of the numerical simulation of this experiment, obtained using the modeling approach in this paper, are shown in Figure 1, where the simulation and experimental results trend in agreement and the simulation results are in better agreement.

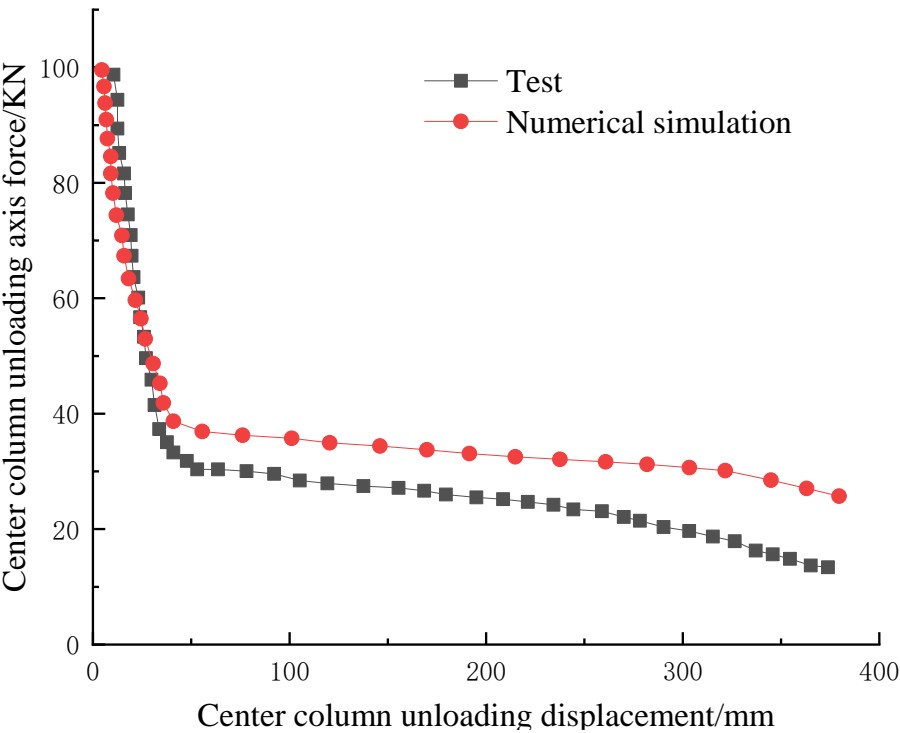

**Figure 1.** Shows the relationship between axial force and unloading displacement.

## 3. Application of Orthogonal Test Method

### 3.1. Finite Element Analysis Steps

(1) Apply gravity, constant load, and live load to the space frame structure; (2) use the modal change command of the life-and-death unit in the ABAQUS interaction module to fail the column element; (3) apply displacement, velocity, etc., to the analysis step of removing the failed column to simulate the initial condition of the transient failure of the column.

#### 3.1.1. Single-Factor Initial Condition Analysis

According to GSA 2003 [2], the column failure time should be less than 0.1 T when performing a nonlinear dynamic analysis of structures. In this paper, 0.1 T was selected as the failure time of the column. The five working conditions of initial vertical speed, 60 mm/s, −30 mm/s, 0 mm/s, 30 mm/s and 60 mm/s, were analyzed under the failure of the bottom side column. The initial velocity was positive upward and negative downward, and the top displacement–time curve of the failed column was obtained, see Figure 2.

It can be seen from Figure 2 that after the failure of the bottom side column, the displacement is 77.88 mm without considering the initial velocity. The maximum displacement of the top of the failed column is 94.78 mm when the initial velocity is up. Furthermore, the maximum displacement of the top of the failed column is 84.46 mm when the initial velocity is down. The results show that the dynamic effect of the initial condition of the initial upward velocity on the structure is amplified, which harms the safety of the structure and will cause the structure to be more seriously damaged, and the greater the initial upward velocity, the greater the vertical displacement of the failed column top.

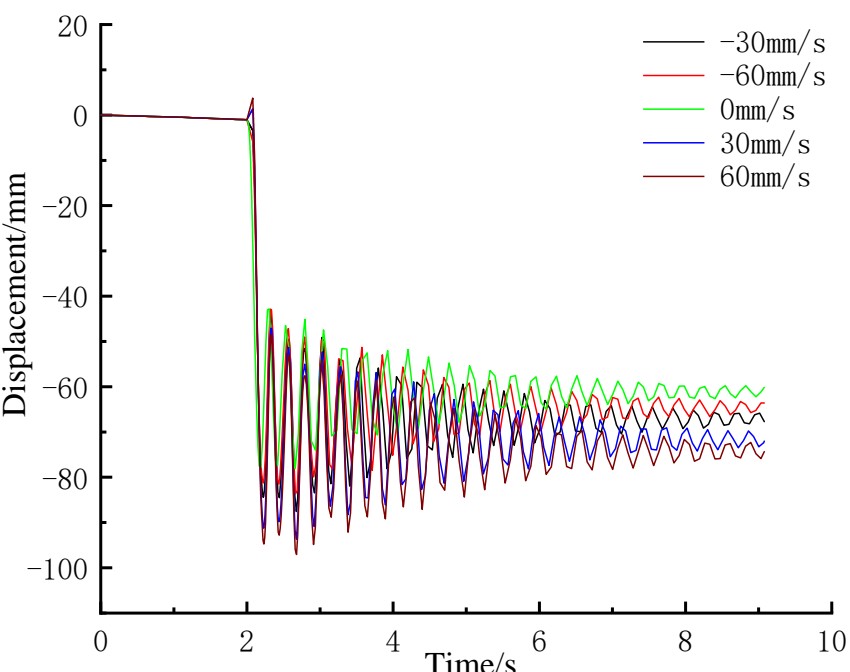

**Figure 2.** Effect of the initial velocity.

### 3.1.2. Effect of Initial Condition Displacement

Similarly, the failure time of the bottom side column is 0.1 T. The five working conditions of initial vertical displacement, −20 mm, −10 mm, 0 mm, 10 mm, and 20 mm, are analyzed in the case of bottom side column failure: the initial displacement is positive upward and negative downward, and the top displacement–time curve of the failed column is obtained; see Figure 3.

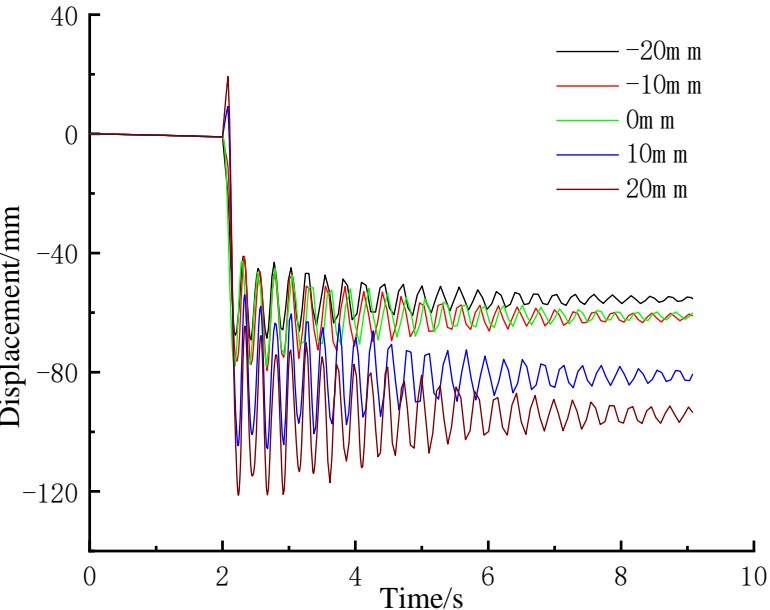

**Figure 3.** Effect of initial displacement.

It can be seen from Figure 3 that when the bottom side column fails, the maximum displacement of the top of the failed column without considering that the initial displacement is 77.88 mm, and the maximum displacement of the top of the failed column is 124.30 mm when the initial displacement is up. The maximum displacement of the top of the failed column is 76.28 mm when the initial displacement is down. Similar to the initial velocity

that exists upward, the initial displacement upward has a more significant impact on the dynamic response of the structure than the initial displacement downward. Unlike the initial downward velocity, the initial downward displacement reduces the maximum displacement at the top of the failed column.

3.1.3. Discussion of the Results

The existence of the initial condition has a more significant impact on the dynamic response of the structure, related to whether the initial upward velocity or the upward initial displacement amplification of the dynamic response of the structure is greater than the initial downward speed and initial downward displacement. The initial upward condition damages the structure's resistance to continuous collapse and tends to cause more significant damage. The upward initial displacement and velocity will cause the structure to produce downward resistance. Its load is also downward, and the structure's downward effective load increases, resulting in more significant vertical displacement at the top of the structural failure column. The initial downward displacement will cause the structure to produce upward resistance, reduce the downward payload of the structure itself, and reduce the top displacement of the failed column. The dynamic response generated by the initial downward velocity is smaller than that caused by the initial upward velocity. However, it does not reduce the vertical displacement of the top of the failed column. The upward resistance may be generated by the initial downward velocity of the structure. The resulting resistance is less than the resistance produced by the initial downward displacement. The initial downward velocity causes the downward payload to be greater than the payload of the initial downward displacement, so the initial downward velocity is greater than the dynamic response generated by the initial downward displacement.

*3.2. Orthogonal Test Method*

3.2.1. Trial Arrangement

In this paper, the three factors selected to affect the dynamic response of the structure are failure time, initial velocity, and initial displacement. The five levels of failure time factor are 0.05 T, 0.075 T, 0.1 T, 0.125 T, and 0.15 T; the five levels of initial velocity factor are −60 mm/s, −30 mm/s, 0 mm/s, 30 mm/s, 60 mm/s; the five levels of initial displacement factors are −20 mm, −10 mm, 0 mm, 10 mm, 20 mm. The combination of orthogonal table L25(53) arrangement factors were used to analyze the structure's dynamic response. The test schemes under the three working conditions of the bottom side, corner, and inner columns are shown in Tables 3–5. The vertical displacement of the top of the failed column under different conditions of the bottom corner, side, and inner columns is shown in Figures 4–6.

**Table 3.** Orthogonal test table when bottom side column fails.

| Factor | Failure Time | Initial Velocity | Initial Displacement |
|--------|--------------|------------------|----------------------|
| K1 | 478.3 | 479.2 | 623.8 |
| K2 | 476 | 471.2 | 521 |
| K3 | 477.4 | 455 | 432 |
| K4 | 464.1 | 477.9 | 399.1 |
| K5 | 462.6 | 475.1 | 382.5 |
| k1 | 95.66 | 95.84 | 124.76 |
| k2 | 95.2 | 94.24 | 104.2 |
| k3 | 95.48 | 91 | 86.4 |
| k4 | 92.82 | 95.58 | 79.82 |
| k5 | 92.52 | 95.02 | 76.5 |
| R | 3.14 | 4.84 | 48.26 |

**Table 4.** Orthogonal test table when bottom corner column fails.

| Factor | Failure Time | Initial Velocity | Initial Displacement |
|---|---|---|---|
| K1 | 425.9 | 426.8 | 547 |
| K2 | 420.3 | 417.5 | 452.6 |
| K3 | 417.4 | 398.2 | 376.1 |
| K4 | 413 | 423 | 354.3 |
| K5 | 407.8 | 418.9 | 354.4 |
| k1 | 85.18 | 85.36 | 109.4 |
| k2 | 84.06 | 83.5 | 90.52 |
| k3 | 93.48 | 79.64 | 75.22 |
| k4 | 82.6 | 84.6 | 70.86 |
| k5 | 81.56 | 83.78 | 70.88 |
| R | 3.62 | 5.72 | 38.54 |

**Table 5.** Orthogonal test table when the bottom inner column fails.

| Factor | Failure Time | Initial Velocity | Initial Displacement |
|---|---|---|---|
| K1 | 618.9 | 615.5 | 787.8 |
| K2 | 617.1 | 614.2 | 681.1 |
| K3 | 618.2 | 587 | 561.3 |
| K4 | 604 | 615.6 | 530.5 |
| K5 | 587.4 | 613.3 | 484.9 |
| k1 | 123.78 | 123.1 | 157.56 |
| k2 | 123.42 | 122.84 | 136.22 |
| k3 | 123.64 | 117.4 | 112.26 |
| k4 | 120.8 | 123.12 | 106.1 |
| k5 | 117.48 | 122.66 | 96.98 |
| R | 6.3 | 5.75 | 60.85 |

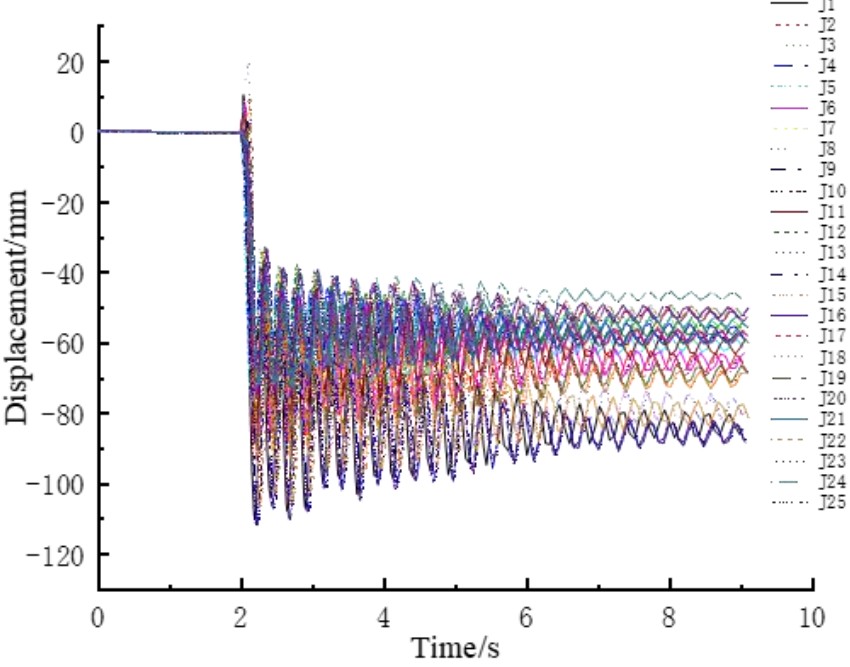

**Figure 4.** Vertical displacement of column top after the failure of bottom corner column.

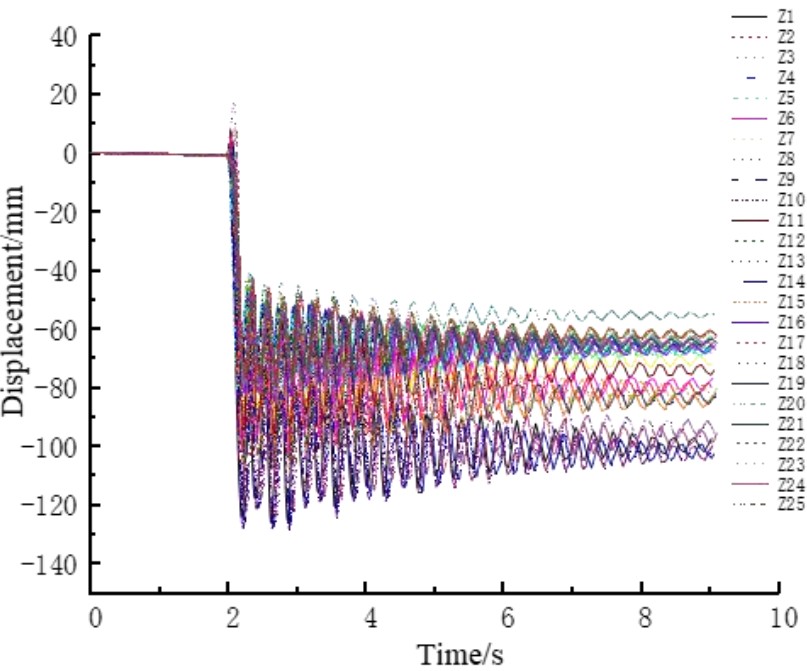

**Figure 5.** Vertical displacement of column top after bottom side column failure.

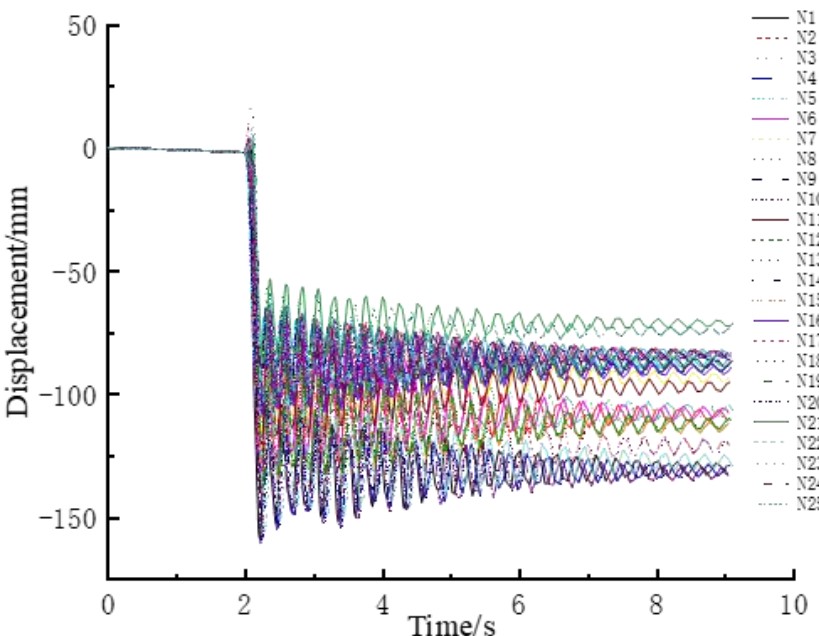

**Figure 6.** Vertical displacement of column top after the failure of bottom inner column.

### 3.2.2. Range Analysis

To obtain the degree of influence of each factor on the dynamic response of the remaining structure, the data will be analyzed using polar difference analysis. The data in the table will be obtained according to Equations (1)–(3). The maximum vertical displacement of the top of the failed column indicates the dynamic response of the structure.

Figure 7 shows that when the initial displacement is positive, the more significant the velocity $k_i$, the greater the average effect of increasing velocity on the dynamic response of the structure. When the displacement is negative, the initial displacement's average effect on the structure's dynamic response decreases slightly from 0 to −10 mm. The average effect on the bottom corner and bottom side columns increases slightly from −10 mm to −20 mm, with a more pronounced increase in the bottom inner column.

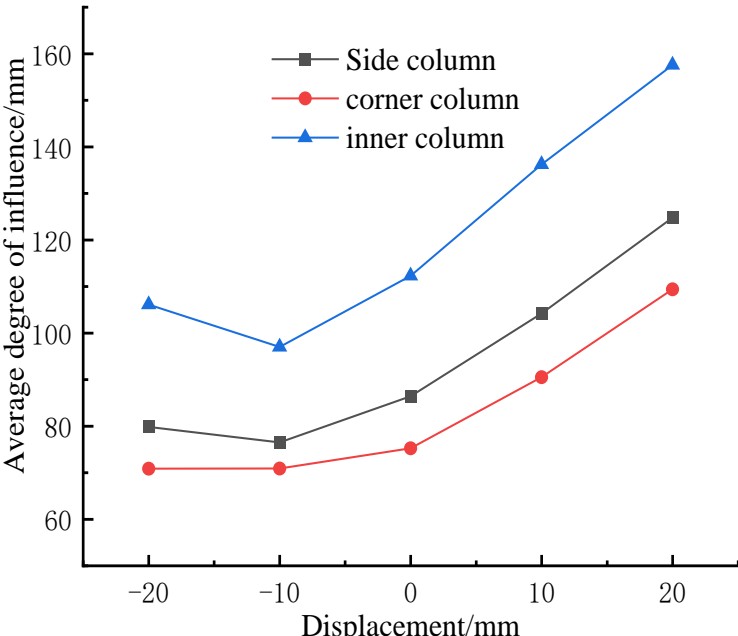

**Figure 7.** Initial displacement.

From Figure 8, when the velocity is up, the average influence degree of the increase in the initial velocity on the dynamic response of the structure tends to be a slight increase. When the direction of the initial rate is downward, the average degree of influence on the dynamic response of the structure increases at the initial speed of 0 to −30 m/s. In contrast, the average degree of influence decreases slightly from −30 to −60 mm/s. The average degrees of influence of the initial velocity on the dynamic response of the structure after the failure of the bottom corner column, bottom side column, and bottom inner column follow the same trend.

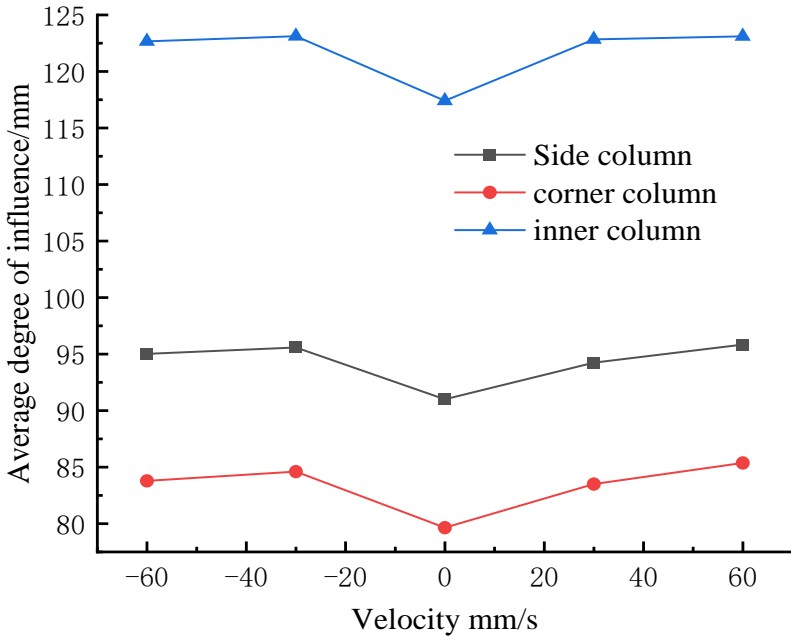

**Figure 8.** Initial velocity.

From Figure 9, when the failure time gradually increases, the average influence of the failure time on the dynamic response of the structure tends to be flat and weakened. The

change in failure time has little effect on the dynamic response of the structure after the failure of the column in the three positions.

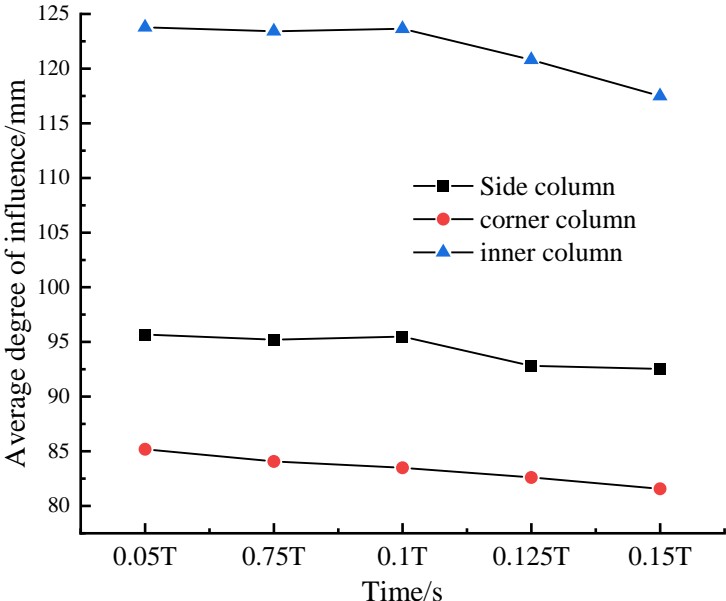

**Figure 9.** Element failure time.

The maximum average degree of influence of each factor to derive the most unfavorable combination affecting the dynamic response of the structure. The maximum average influence degree of initial displacement is 20 mm, the maximum average influence degree of initial velocity is 60 mm/s, and the maximum average influence degree of failure time is 0.05 T. The combination that produces the most significant structural dynamic response is the failure time of 0.05 T; the initial speed is 60 mm/s, and the initial displacement is 20 mm. In the case of failure of all three types of columns at the bottom corner, side, and inner columns, this combination produces the greatest dynamic response to the structure. This is the most detrimental to the structure's resistance to continuous collapse. Figures 10–12 are the most unfavorable combination displacement deformation diagrams under the failure of the three columns.

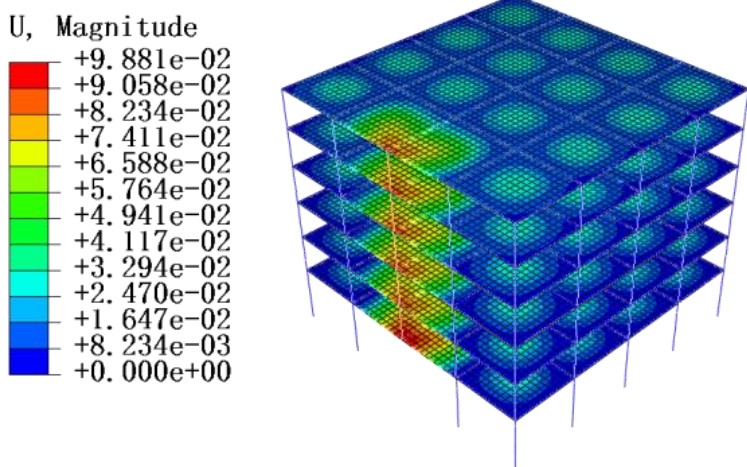

**Figure 10.** The most unfavorable combination displacement diagram of the bottom side column.

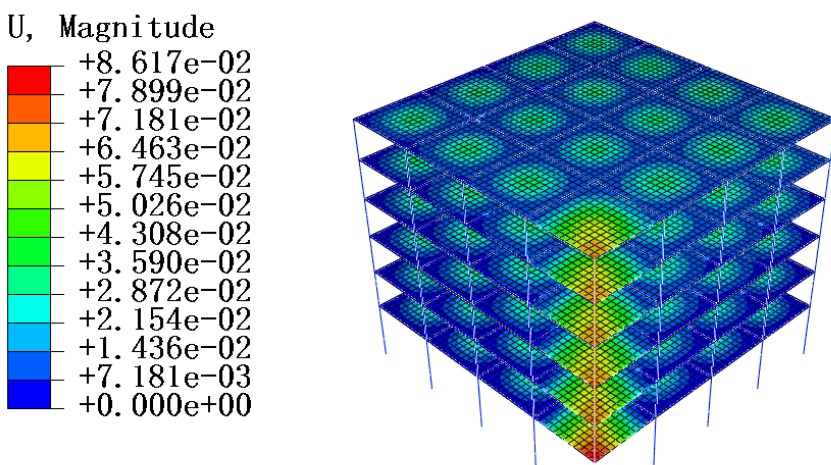

**Figure 11.** The most unfavorable combination displacement diagram of the bottom corner column.

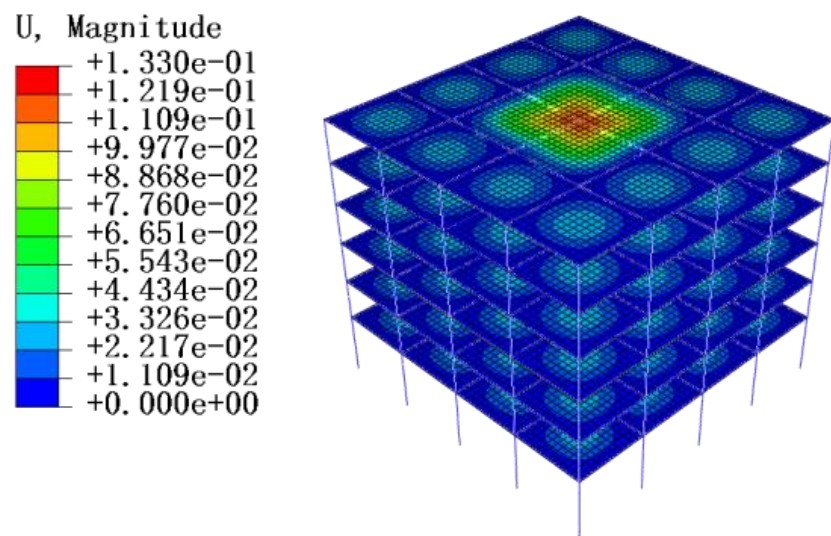

**Figure 12.** The most unfavorable combination displacement diagram of the bottom inner column.

The extreme difference values R in Tables 1–3 were obtained by Equation (3): the failure time R1 of the bottom side column is 3.14; the initial velocity R2 is 4.84; the initial displacement R3 is 48.2; the bottom corner column R1 is 3.62; R2 is 5.72; R3 is 38.54. The bottom inner column R1 is 6.3, R2 is 5.75, and R3 is 60.85. The failure time and initial velocity have little influence on the dynamic response of the structure under the three failure conditions of the bottom corner, side, and inner column. Furthermore, the time interval selected for the failure time was small, resulting in less influence on the factor of failure time. However, the initial displacement has a more significant influence on the dynamic response of the structure under the three failure conditions of the bottom corner column, side column, and inner column. Section 3.2 shows that when the initial velocity is down, the dynamic response to the structure cannot be weakened. However, this is only smaller than the dynamic response generated by the initial upward velocity, and the initial downward displacement can weaken the dynamic response of the structure. Compared to the initial velocity, the initial downward displacement can attenuate the dynamic response of the structure, and the initial velocity has less influence than the initial displacement.

## 4. Conclusions

In this study, a single factor of initial velocity and initial displacement of the remaining structure was considered for dynamic structural analysis under the conditions of failure

of the side columns. Furthermore, a combined analysis of three factors of failure time, initial velocity, and initial displacement under the conditions of failure of each of the three columns, obtained using the orthogonal test method, led to the following conclusions:

(1)    When the column failure time is close to 0.1 T, the failure time has little influence on the dynamic response of the structure.

(2)    By comparing the dynamic response of the structure under different initial displacement and velocity conditions, it is found that the progressive collapse analysis, without considering the initial condition, underestimates the influence of the initial condition on the structural deformation. The initial upward velocity and initial upward displacement will amplify the dynamic response of the structure. This will cause more severe damage to the structure, which is unfavorable to the progressive collapse resistance of the structure. The initial downward displacement and initial downward velocity are beneficial to the progressive collapse resistance of the structure, and the initial downward displacement will weaken the dynamic response of the structure.

(3)    In all three column failure cases, the most unfavorable combinations of structural dynamic response were 0.05 T failure time, 20 mm initial displacement, and 60 mm/s initial velocity. The failure time and initial velocity have little influence on the dynamic response of the structure, and the initial displacement has a more significant influence on the structure. At present, there are few studies considering the initial condition of the residual structure and the initial condition of the residual structure that affects the resistance of the structure to progressive collapse. Considering the initial condition of the remaining structure can more accurately analyze the progressive collapse resistance of the structure.

## 5. Further Research

The following points may be considered for further studies:

1.    The dynamic responses of other structures and columns of different floors after failure should be studied, considering the initial condition of the remaining structure.

2.    The influence of the presence of floor slabs on the dynamic response of the structure under the initial condition of the remaining structure should be considered.

3.    The progressive collapse resistance of long-span space structures can be studied by an orthogonal test.

**Author Contributions:** Conceptualization, methodology, and review, C.K. and J.J.; numerical models, validation, and original draft preparation, X.L. All authors have read and agreed to the published version of the manuscript.

**Funding:** This research received no external funding, and the source of APC is self-raised.

**Institutional Review Board Statement:** Not applicable.

**Informed Consent Statement:** Not applicable.

**Data Availability Statement:** All the analysis data in this paper are from numerical simulation, and I guarantee the authenticity and validity of the data.

**Conflicts of Interest:** The authors declare no conflict of interest.

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
