# Peer review of "Dynamic Response Analysis of RC Frame against Progressive Collapse Based on Orthogonal Test"

_applsci, doi:10.3390/app13074317_

Round 1

Reviewer 1 Report

It is necessary to expand the list of references, as well as the review of the papers in the first part of the paper (Introduction).

It is extremely important to edit the text so that it is much more understandable to the readers and in the spirit of the English language, including the captions below Figures 2 and 3 which should be also corrected to make them more understandable to the readers.

The title of chapter 3.2.3 would be more suitable to be corrected to "Discussion of the results" or "Results discussion".

The conclusions should be expanded.

Reviewer 2 Report

The research presented in the article entitled “Dynamic Response Analysis of RC Frame Against Progressive Collapse Based on Orthogonal Test” addresses to the degree of influence of different factors on the structural dynamic response. The article ensures very complex and useful data about an innovative material for using in a very different environment than usual. It is a very interesting study, with a high focus on the future possibilities.

The Abstract presents a clear and comprehensive statement of the study described in the paper.

The Introduction and the entire paper provide enough literature references about the article’s subject and for the analysis performing.

The materials and methods are in detail described. The results are in detail presented, clearly highlighted and discussed, and the conclusions summarize them properly.

Reviewer 3 Report

Check the attached file please

Round 2

Reviewer 1 Report

The paper has been significantly improved and I suggest that it should be published in the present form.

Reviewer 3 Report

No more comments